# Analysing fast food consumption behaviour patterns: The case of Jordan

**Noura Abu Asab** [ORCID]**[1]\***, **Randah Barakat[2]**

**1** Department of Business Economics, School of Business, The University of Jordan, Amman, Jordan,
**2** Department of Counselling and Mental Health, School of Educational Sciences, The World Islamic Sciences and Education University, Amman, Jordan

\* n.abuasab@ju.edu.jo

## Abstract

This study aimed to examine the factors influencing fast food consumption among a Jordanian population by utilizing the extended theory of planned behavior as its theoretical framework. The research employed theory-related scales, income, BMI, ability to consume (intention), indirect intention, and a retrospective measure of fast-food consumption within a cross-sectional sample of 408 university students aged 18–23 years. Structural equation modeling was used to analyze the determinants of consumption. The findings indicated that the self-reported intention measure failed to mediate the attitude-behavior relationship. However, the use of projective intention (indirect) measures revealed that personal factors such as attitude, subjective norms, and self-identification significantly explained over 50% of the variation in the intention to consume fast food, suggesting that participants often underestimated their intention to consume fast food. Furthermore, the study identified available personal funds, perceived and actual behavioral control, and BMI as significant external predictors of fast-food consumption. The originality of this study lies in its contribution to understanding the growing preference for fast food in the Middle East, particularly in Jordan. It introduces a theoretical model that employs projection-based assessments to implicitly gauge intentions and explores the relationship between income, BMI, and consumption behavior. The practical implications of these findings underscore the importance of key psychosocial elements in developing and implementing preventive strategies aimed at promoting healthy eating behaviors among university students.

## 1. Introduction

Fast-food consumption has become increasingly prevalent among students and young adults, reflecting a significant shift in lifestyle and dietary habits. This trend can be attributed to several factors, including the influence of Western food cultures portrayed in the media, internet marketing and advertising [1], the rise of multinational food chains in the 1990s [2] and rapid urbanization, which has altered living patterns [3–5].

**Data availability statement:** All relevant data are within the paper and its Supporting Information files.

**Funding:** The author(s) received no specific funding for this work.

**Competing interests:** The authors have declared that no competing interests exist.

Additionally, factors such as increased income, hectic lifestyles, the convenience of home delivery with food safety assurance, and the emergence of new brands offering various specialties have contributed to the rise in fast-food consumption. These elements influence young adults' eating habits, particularly during periods of significant independence in food planning, shopping, and preparation [6], potentially leading to poor nutritional choices [7].

Fast-food companies have also capitalized on changing demographics and customer psychographics by employing various advertising methods that primarily target children and young adults. Unfortunately, the proliferation and density of fast-food outlets are linked to the global obesity epidemic [8–12], with fast-food consumption potentially contributing to rising obesity rates [13].

Recently, there has been a notable shift in the dietary habits of individuals in the Middle East, particularly Jordan. This shift has resulted in an increase in the consumption of unhealthy fats, sugars, and processed carbohydrates, exacerbated by the proliferation of American food franchises. According to official data from various institutions, a significant proportion of Jordanians exhibit unhealthy eating habits, with approximately 84% of the population affected [14]. Moreover, the prevalence of overweight individuals increased from 60.6% to 69.6% between 2000 and 2016, according to the World Health Organization [15]. Consequently, there are growing concerns regarding high BMI, dietary hazards, elevated systolic blood pressure, and other chronic diseases in Jordan [16]. Therefore, it is crucial to identify the behavioral and economic factors associated with increased fast-food consumption.

This study is grounded in the assumption that fast-food consumption among young adults in Jordan is influenced by a combination of behavioral intentions, perceived norms, personal identity, and financial and physiological factors. While existing research has explored dietary behavior broadly, few studies have examined these drivers using an extended Theory of Planned Behavior (TPB) in Middle Eastern contexts. The focus on university students reflects a critical life stage where individuals gain autonomy over their food choices, making them particularly vulnerable to unhealthy dietary habits. The University of Jordan, one of the country's largest and most demographically diverse institutions, offers an appropriate setting for this investigation.

The primary objective of this study is to examine how the extended TPB constructs, along with income (personal funds), BMI, and fear of negative evaluation, shape both intention and actual fast-food consumption behavior. A key methodological contribution lies in the use of a projection-based measure of intention to reduce self-report bias.

By targeting a relatively underexplored population and integrating extended behavioral predictors, this study contributes to the literature by offering a context-specific and theory-informed model that can inform culturally tailored health promotion strategies in Jordan and similar settings.

The article is structured as follows: Section Two presents the theoretical background, Section Three outlines the research methodology, ethical consideration

and research design, Section Four displays the results obtained, Section Five discusses the results, and Section Six concludes.

## 2. Theoretical background

The Theory of Reasoned Action (TRA), proposed by [17], holds significant relevance in the realm of food behavior. This theory posits that behavior is guided by intention, which is shaped by an individual's attitude towards the behavior and subjective norms. Specifically, the TRA suggests that normative beliefs and willingness determine subjective norms, while behavioral values impact attitudes towards the behavior. Building on this, Ajzen [18] introduced the concept of perceived behavioral control to the TRA, leading to the development of the TPB [19]. The TPB asserts that most deliberate behavior is rational and guided by individuals' objectives, with intentions mediating the relationship between attitudes and behavior [20].

The TPB further posits that specific factors directly impact behavioral intentions, which in turn affect behavior [21]. Attitudes towards a behavior are shaped by an individual's beliefs about the consequences and their perception of these consequences. Additionally, subjective norms are formed by an individual's belief about what others expect from them and their motivation to fulfil these expectations. These perceptions can be divided into descriptive norms, describing how others around an individual behave, and injunctive norms, referring to the anticipated or perceived likelihood of approval or disapproval from a specific individual or group [22]. Parents play a crucial role in introducing their children to new foods, establishing dietary standards, and ensuring an adequate food supply at home. As individuals age, peers become increasingly important in various social aspects [23–25].

A crucial aspect of the TPB is "perceived behavioral control" (PBC), which measures an individual's confidence in executing a specific behavior, influenced by factors such as opportunities, resources, skills, and the impact of beliefs. The TPB model also recognizes ABC, which refers to external circumstances influencing behavior but not immediately controllable by the individual. In consumer behavior, the perceived value of a product is defined as the difference between the perceived and expected benefits to the consumer [26]. This concept involves a trade-off between what the consumer gives up and what they receive in exchange for purchasing a product [27] and is characterized as preferential, perceptual, and cognitive-affective in nature [28]. Numerous studies have demonstrated that a product's perceived value significantly influences consumers' purchasing decisions (e.g., [29–31]). Dodds and Monroe [32] argued that consumers are more likely to purchase a product they perceive as valuable, combining transaction and acquisition utility, making it an essential factor influencing consumer purchasing intentions [33]. Despite this, high prices can be a significant barrier to purchasing food products, especially for low-income individuals. High prices reduce consumers' ability to buy a product, causing uncertainty and difficulty in making purchase decisions [34]. Various studies confirm that pricing is a key consideration for many customers when making purchasing decisions [35].

The concept of purchase intention is widely recognized in consumer behavior, with researchers such as [36–40] emphasizing its importance in predicting the likelihood of a specific purchase. Numerous studies have demonstrated that the TPB can explain a considerable portion of the variation in intention and behavior [41]. In recent years, this theory has been extensively applied in the literature on food consumption, offering valuable insights into the factors influencing consumption behavior.

Research has dedicated significant effort to understanding individual differences in factors that may impact, enhance, or diminish the variables included in the TPB. Consequently, Dunn et al. [20] proposed additional constructs encompassing other factors previously suggested to affect food consumption behaviors. This includes the inclination to identify as a "healthy eater," as indicated by [42], which possess the potential to equal cognitive responses. Studies on the latter suggest that individuals often prioritize short-term benefits, represented by affective actions such as taste and convenience, over potential long-term costs, represented by cumulative health risks associated with food choices. Those aware of the long-term health risks would disregard the immediate reward of taste and convenience from fast food intake, as proposed

by [43,44]. Conversely, individuals who give little consideration to long-term effects may opt for short-term rewards. Increasing consumer awareness of food origins, spurred by recent food scarcity, environmental concerns, animal welfare, and illicit practices, has led to growing demand for organic, vegetarian, and additive-free food [45].

Another relevant factor is the Fear of Negative Evaluation (FNE), where individuals are concerned about others' opinions of them [46–49]. This phenomenon can manifest in various contexts, including food consumption [20,50]. Additionally, research indicates that individuals from low-income or lower educational backgrounds are typically exposed to environments with a high prevalence of high-energy meals [51–53]. Studies also indicate that preparing healthy food at home is more cost-effective than consuming unhealthy food at fast food restaurants [54]. However, some researchers argue that healthy meals are more expensive than unhealthy ones, leading lower-income individuals to spend less time on unhealthy foods [23].

While some studies have found that the Theory of Planned Behavior does not adequately explain behavior towards fast food consumption, many researchers argue that it is essential in explaining food consumption behaviors [23,45,55–59]. The present study aimed to investigate the determinants of fast-food consumption among students at the University of Jordan, using the extended TPB and incorporating income (personal funds) and BMI measures as predictors for consumption behavior.

## 3. Study design, methodology, and ethical consideration

The research design for this study was a cross-sectional survey. This design allowed for the examination of relationships between variables as proposed by the TPB and the extended model. Students were selected as the population for this study due to their unique dietary habits and susceptibility to fast-food consumption, making them an ideal group for understanding the determinants of such behavior.

The sample size for the study was determined based on the population of students at the University of Jordan and the need for sufficient statistical power to detect meaningful effects. A sample size of 408 students was deemed appropriate, ensuring a representative sample while allowing for robust statistical analysis. The respondents were selected using stratified random sampling to ensure diverse representation across different schools and academic levels. The University of Jordan was selected due to its large student population, academic diversity, and geographic centrality, which make it a strong proxy for broader student populations in the country. While it may not represent all Jordanian students, its heterogeneity offers a valuable cross-section for initial analysis. The observed gender imbalance (75% female) is consistent with national enrollment patterns, particularly in the social sciences and humanities.

The student population at the University of Jordan was divided into strata based on two primary criteria: School and academic level. To ensure that each stratum is proportionally represented in the sample, the number of students selected from each school and academic level was proportional to the size of the respective strata in the overall student population. Within each stratum, students were randomly selected. This ensured that the selection process was unbiased and each student within the stratum had an equal chance of being selected.

Selected students were approached to participate in the study. They were provided with the survey questionnaire and informed about the purpose of the study, ensuring voluntary participation and informed consent. Data collection involved a structured questionnaire, divided into two sections. The first section focused on demographic factors and income thresholds, while the second addressed behavioral questions based on the TPB, including attitudes towards behavior, subjective norms, perceived behavioral control, and actual behavioral control. Similar to [20] study, additional factors such as belief strength, consideration of future consequences, and self-identification were included, accounting for possible interaction effects between these factors.

Structural Equation Modeling (SEM), which includes confirmatory factor analysis and path analysis, was conducted to simultaneously analyze multivariate causal relationships based on the cross-sectional survey. The survey incorporated variables related to students' available income (financial access or flexibility – students' available personal funds, including

family support, stipends, scholarships, and part-time work), height, and weight (to calculate BMI). Table 1 presents the comparison of the demographic characteristics of the groups.

The data were analyzed using SPSS 27 and AMOS 26 software. Table 1 summarizes the mean responses for all variables. The Pearson correlation test was used to assess the relationship between the TPB elements. Bootstrapping was employed to correct for non-normality. The TPB's internal consistency was tested using Cronbach's alpha, although its application is limited to unidimensional construct items and its sensitivity varies with the length of the questionnaire per construct [60]. Constructs with fewer than 10 questions exhibiting a Cronbach's alpha value over 0.5 indicated good reliability [61].

Different criteria were used to assess model fit, including the relative chi-square index, Comparative Fit Index (CFI) [0 (poor fit), 1 (very good fit)], which compares the absolute fit of the specified model to the absolute fit of the independent model, Adjusted Goodness of Fit (AGFI) [0 (no fit), 1 (perfect fit)], and Root Mean Square Error of Approximation (RMSEA) [<0.05 (good model fit); <0.10 (marginally good model fit)]. "Due to the sensitivity of the Chi-square test to large sample sizes, the relative chi-square index ($\chi^2/df$)- Chi-square divided by degrees of freedom- is employed. Value ≤ 3 is considered acceptable-fit [62], while value ≤ 5 indicates a reasonable fit [63]. RMSEA values between 0,08 and 0,1 are considered borderline, values ranging from 0,05 to 0,08 are considered acceptable. MacCallum et al. [64] proposed that an RMSEA value of.05 -.08 is satisfactory. PGFI value of 0,5 or greater indicates an acceptable fit. AGFI value of 0,80 or greater indicates an acceptable fit [65]. Kline [62] suggests that values between 0,90 and 0,95 are acceptable fit and Values above 0,95 are excellent fit. RMR ≤ 0.07 = acceptable fit [66]. A value CFI ≥ 0,90 indicates a perfect fit [67]".

**Table 1. Participants' characteristics and types of food consumed.**

| Variable | Variance levels | Frequency (%) |
|---|---|---|
| **Gender** | Male | 100(24.5) |
| | Female | 308(75.5) |
| **living with** | Family | 391(95.8) |
| | Study/work colleagues | 2(0.5) |
| | Alone | 15(3.7) |
| **Income** | <100 | 213(52.2) |
| | 100-199 | 93(22.8) |
| | 200-299 | 42(10.3) |
| | 300-350 | 19(4.7) |
| | 351-450 | 10(2.5) |
| | >450 | 31(7.6) |
| **BMI** | Underweight (less than 18.5) | 26(6.4) |
| | Normal (between 18.5 and 24.9) | 274(67.2) |
| | Overweight (between 25 and 29.9) | 84(20.6) |
| | Obese (more than 29.9) | 24(5.9) |
| **Which of the following foods do you consume the most?** | Shawarma and crispy chicken | 249(60.9) |
| | Fried foods (Potatoes, Chicken, Falafel, Donut) | 217(53.1) |
| | Processed meat (Burger, Sausage) | 125(30.6) |
| | Fast -prepared food | 117(28.6) |
| | Pastries of all kinds | 97(23.7) |
| | All types | 77(18.8) |

The questionnaire used in this investigation was authorized by the University of Jordan's Deanship of Scientific Research under approval decision number (37–2023). Before distributing the questionnaire, the objectives of the study were clearly communicated to the students to ensure they understood the purpose and nature of the research. Informed consent was obtained by requiring participants to indicate their approval by checking a box labeled "I approve to participate." The online questionnaire was administered in the presence of researchers to provide participants the opportunity to ask questions and receive clarification as needed. Students were recruited from the University of Jordan between 22/10/2023 and 18/12/2023.

A total of 408 participants provided feedback for the survey, resulting in a response rate of 93% of a total of 439 students. The majority of the participants (75%) were female. The findings revealed that almost all the respondents (96%) resided with their families and were unmarried. Regarding their monthly available income amounts, which serve as a measure of student's financial access or flexibility, over half (50%) reported receiving less than $140, while around one-quarter (25%) stated earning between $140 and $280 per month. Approximately 8% disclosed receiving more than USD 630 per month.

The data were collected from 408 observations gathered from students who completed a Likert-scale survey consisting of response options ranging from "strongly disagree" to "strongly agree." The scale also included "disagree," "neutral," and "agree" as response options. As a measure of actual behavior, students were requested to identify the frequency category for each food item in terms of their normal intake, with the scale ranging from "never or less than once-twice per month" to "one or more times per day."

The questionnaire presented several fast-food options for participants to select from, including burgers (similar to McDonalds), hot dogs (similar to Hot Dog of Thrones), French Fries (similar to McDonalds), Fried Chicken (similar to KFC), Falafel (similar to Abu Jbara), Donut/Arabic Sweets (similar to Crispy crème/Habibah), Ready to cook meals (similar to Maggi soups, Indomie), and Shawarma (similar to Aldayaa). Participants were given the freedom to choose one option, all options together, or multiple choices that they preferred. According to the responses of the students, most of the fast food consumed was Shawarma or Fried Chicken (61%), followed by fried food (53%) and Burgers (30.6%).

The survey also inquired about the monthly income of the students, which ranged from less than 100 JD (140$) to over 450 JD (634$). In this study, the term "income" refers to students' available personal funds, which may include financial support from family, university scholarships, part-time jobs, or stipends. This measure captures the disposable amount of money accessible for personal spending, rather than earned income or full financial independence. While not a perfect proxy for economic autonomy, it serves as a contextual variable reflecting the financial flexibility that may enable or constrain fast-food consumption. The BMI of each student was calculated based on the height and weight information provided in the questionnaire. Despite the absence of any significant demographic differences among the respondents, individuals with a higher income were more likely to be overweight.

Two measures were employed to ascertain self-reported fast-food consumption intentions. The first measure asked participants to evaluate, on a scale of 1–5, the likelihood of consuming fast food regularly over the following month based on their lifestyle and taste preferences. The second measure assessed readiness to consume fast food by inquiring about their inclination to do so within the next month, using a scale ranging from 1 to 5. The internal consistency reliability of these two questions was determined using the Cronbach's alpha coefficient, which yielded a value of 0.79, indicating good consistency. Additionally, an index comprising six facilitating constructs regarding perceptions of fast-food consumption and attention to health and environmental consequences was constructed. The key predictors of intention were mainly derived from the extended TPB.

In addition, the questionnaire was developed in line with the theoretical framework of the TPB, based on previous studies such as [20]. Rather than adopting a fully standardized instrument, the items were adapted to reflect the cultural context, dietary habits, and linguistic norms of Jordanian university students. These adaptations aimed to improve the clarity, relevance, and interpretability of the questions. The instrument was pilot tested and revised accordingly, and internal

consistency, measured using Cronbach's alpha, showed acceptable to strong reliability across constructs ($\alpha = 0.60$–$0.86$). While not based on a single validated tool, this approach ensured theoretical consistency while enhancing contextual appropriateness; a strategy frequently employed in the TPB-based behavioral studies conducted in diverse settings.

### 3.1 Construct of the TPB

The principal factors that predicted intention and behavior, namely, attitude, subjective norms, perceived behavioral control, and actual behavioral control, are outlined below.

**3.1.1 Attitude.** Thirteen items were used to assess the cognitive and affective attitudes. Participants were asked to rate their cognitive attitude by providing their opinions about fast food using the sentence "I think, eating fast food is..." followed by five adjectives, which were rated on a 5-point Likert scale. The adjectives were beneficial, quick, convenient, unpleasant, and cheap, and their scores were combined to form a composite index for this type of attitude. Affective attitudes were measured using nine adjectives presented in the question "I feel that eating fast food is..." followed by nine adjectives rated on a 5-point Likert scale, including happy, guilty-free, energetic, gratified, worried-free, enthusiastic, enticed, and capable. The scores from these responses were averaged to form a composite index that reflected affective attitudes. Both cognitive and affective attitudes were equally weighted in the calculation of the overall assessment variable, which represented the overall attitude toward fast-food consumption. The reliability of this variable was determined using Cronbach's alpha, which was 0.80. The alpha value for cognitive attitude was 0.65, while the alpha value for affective attitude was 0.75.

**3.1.2 Subjective norms (SN).** Four items were used to assess normative beliefs, evaluating both injunctive norms (the beliefs of others) and descriptive norms (the actions of others). Each item is rated on a 5-point Likert scale. The two injunctive items comprised statements such as "Individuals important to me deem it necessary to consume fast food on a regular basis" and "My close-knit circle anticipates that I frequently consume fast food." The two descriptive items were "Those in my inner circle frequently indulged in fast foods" and "Individuals whose opinions carry weight with me, such as university professors or social media influencers, frequently consume fast food."". To derive an overall assessment of Social Norms, the composite variable combined the scores from both scales. Cronbach's alpha, with a value of 0.65, served as a marginally reliable indicator of the consistency of the SN scale.

**3.1.3 Perceived behavioral control (PBC).** Four items were used to assess perceptions of control and self-efficacy. Perception of control was evaluated using two 5-point Likert scale items. The first item assessed was "I could control the number of times I will consume fast food in the upcoming month," while the second item was "The frequency at which I will eat fast food within the next month is primarily under my control." Self-efficacy was measured using two items rated on a 5-point Likert scale. These items were "I cannot avoid eating fast food regularly over the next month" and "If I choose to, I could prevent myself from consuming fast food regularly over the next month.".

To create an overall measure of PBC, the scores from both the self-efficacy and behavioral control scales were combined, resulting in a composite variable for PBC. The Cronbach's alpha for this scale was 0.60.

**3.1.4 Actual behavioural control (ABC).** To evaluate the limitations of life or factors that affect an individual's ability to control fast- food consumption, two items were introduced. These were: 'Due to my business or illness-disability, I am unable to prepare food for myself "and 'As there are numerous fast-food outlets near me, I find it more convenient not to cook my own meals'. Additionally, 'My financial ability allows me to eat fast food regularly' was also assessed. The evaluation was conducted using a 5-point Likert scale. However, no evaluation of internal consistency was carried out because these two items were not necessarily correlated with each other [20].

**3.1.5 Brief fear of negative evaluation (FNE) scale.** The four-item measure developed by [46] was evaluated using a 5-point Likert scale. The items included in the measure were: "Concerned about others' opinions of me," "Fearful of rejection," "Worried about the impression I make," and "Anxious about saying or doing something wrong." The internal consistency of the measure was demonstrated by a high Cronbach's alpha value of 0.86.

**3.1.6 Self-identification scale.** The implementation of the 4-item scale, proposed by [42], was conducted to assess an individual's level of self-identification as a healthy eater. This was done using a 5-point Likert scale, with the four items included: I consider myself a healthy eater, I am concerned with eating healthily, I care about the impact of food choice on my health, and 'I enjoy eating.' Additionally, An additional item was added to consider the environmental consequences associated with food consumption, which was phrased as follows: "I consider how my dietary choices affect the environment." The internal consistency of the scale was evaluated using Cronbach's alpha, which yielded a value of.70. "Consideration of future consequences index was dropped as its Cronbach's alpha was low due to low number of constructs included."

**3.1.7 Interaction effects.** To examine the potential influence of a subjective norm-FNE interaction, centered values for both variables were computed using a method by subtracting the individual scores from the mean score [20,68]. This involved creating a new interaction variable, denoted as FNE SN, by multiplying the centered scores.

The following hypotheses have been tested considering the theoretical and empirical debates in the literature:

Hypothesis 1a (H1a): Affective attitudes are positively related to behavioral intention to consume fast food.

Hypothesis 1b (H1b): Cognitive attitudes are positively related to behavioral intention to consume fast food.

Hypothesis 1c (H1c): Cognitive attitudes are stronger predictors of the behavioral intention to consume fast food than affective attitudes.

Hypothesis 1d (H1d): Affective attitudes are stronger predictors of the behavior to consume fast food than affective attitudes.

Hypothesis 2a (H2a): Injunctive norms are positively related to behavioral intention to consume fast food.

Hypothesis 2b (H2b): Descriptive norms are positively related to behavioral intention to consume fast food.

Hypothesis 2c (H2c): The injunctive and the descriptive norms are equally strong predictors of the behavioral intention to consume fast food.

Hypothesis 3a (H3a): Perception of control is negatively related to the behavior to consume fast food.

Hypothesis 3b (H3b): Self-efficacy is negatively related to the intention to consume fast food. Hypothesis 3c (H3c): Perception of control is a stronger predictor of the intention to consume fast food than self-efficacy.

Hypothesis 4 (H4): Self-identification as a healthy eater is negatively related to the behavioral intention to consume fast food.

Hypothesis 4 (H4): Income is positively related to the behavior to consume fast food.

Hypothesis 4 (H4): BMI is positively related to the behavior to consume fast food.

The hypotheses developed above are grounded in an extended conceptualization of the TPB, which incorporates both traditional constructs (attitude, subjective norms, and perceived behavioral control) and additional factors such as self-identification, fear of negative evaluation, and external predictors like income and BMI. This theoretical model positions projective intention, rather than self-reported intention, as the key mediator between psychosocial predictors and

 

fast-food consumption behavior. Fig 1 presents the conceptual framework used in this study, illustrating the extended TPB and external predictors. The model also features a projection-based measure of intention, which replaces direct self-reports to reduce social desirability bias. The arrows in the Figure represent both hypothesized and empirically supported pathways, showing how each variable is expected to influence intention and behavior.

## 4. Results

According to the research sample, 27% of individuals consumed fast food on a weekly basis, whereas 26% consumed fast food two to three times per month. The majority of students (approximately 65%) experienced positive emotions, such as happiness, excitement, and attraction towards fast food consumption, which is perceived as time-saving and effortless. However, an overwhelming majority (87%) did not consider fast food cheap. Notably, most participants believed that their good health status allowed them to consume fast food without any potential risks. A summary of the demographic characteristics of the students in this study and the types of fast food they consume is presented in Table 1. Descriptive statistics, including Cronbach's alpha, the mean score, and standard deviation (SD), are presented in Table 2.

The bivariate correlations among self-reported intentions, constructed intentions, attitudes, subjective norms, PBC (excluding ABC), self-identification, and behavior related to fast-food intake by Jordanian students are presented in Table 3. The TPB variables were significantly associated with an increased desire to consume fast food, except for PBC and ABC (P < 0.001). In contrast, fast-food consumption was negatively correlated with PBC and ABC. No significant correlations were found between income or BMI and intention to engage in this behavior. The study revealed a positive

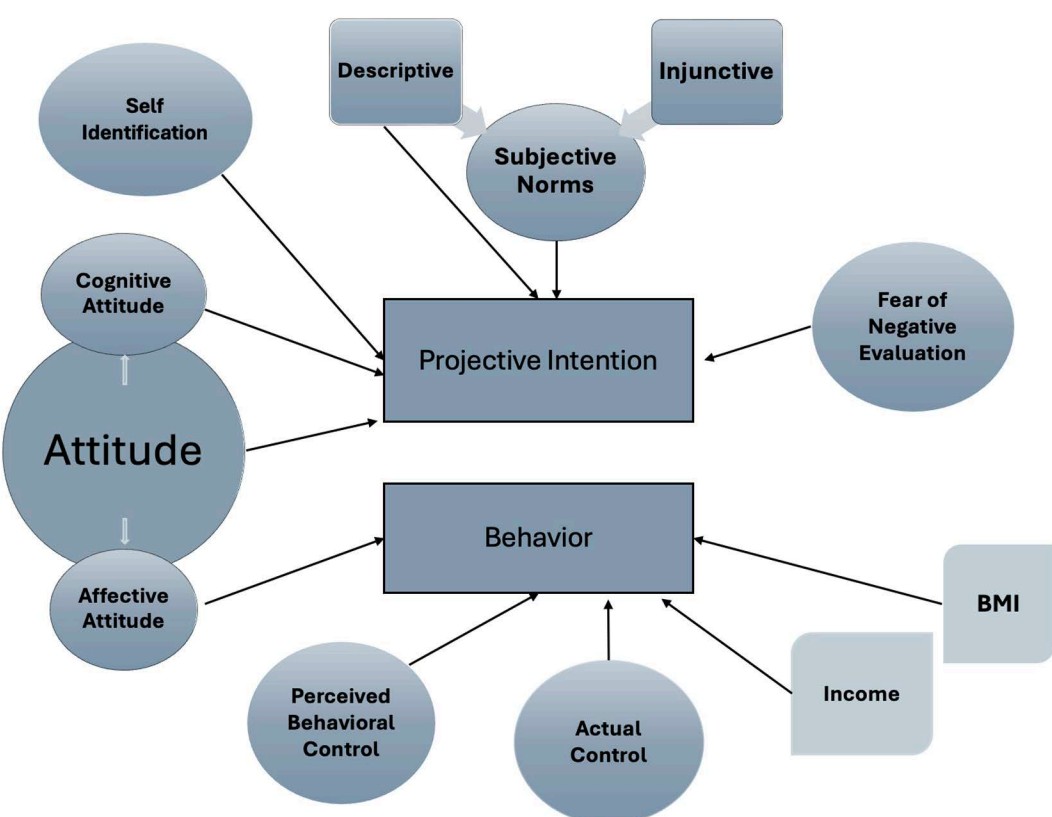

**Fig 1. Conceptual framework of the extended Theory of Planned Behavior (TPB) used in the study.**

**Table 2.** Descriptive statistics; Cronbach's α, mean score, and standard deviation (SD).

| Construct items | Mean (SD) |
|---|---|
| **Cognitive Attitude (α=0.65)** | |
| I think that fast food is not good for your health. | 3.77 (1.035) |
| I think that fast food saves time. | 3.69 (1.051) |
| I think that fast food is easy and convenient. | 3.69 (1.055) |
| I think that fast food is fun. | 3.88 (1.043) |
| I think that fast food is cheap. | 3.64 (0.964) |
| **Affective Attitude (α=0.75)** | |
| I feel happy when eating fast food. | 3.84(0.972) |
| I don't feel guilty when eating fast food. | 3.08(1.276) |
| I feel energized when eating fast food. | 2.47(1.166) |
| I feel full when eating fast food. | 4.25 (0.919) |
| I don't feel anxious when eating fast food. | 3.09(1.325) |
| I feel excited when eating fast food. | 3.62(1.190) |
| I feel attracted to eating fast food. | 3.79(1.100) |
| I feel that my health qualifies me to eat fast food. | 3.43(1.281) |
| ***Subjective Norms* (α=0.65)** | |
| ***Injunctive* Norms** | |
| Most people who are important to me think that I should eat fast food regularly. | 2.69(1.191) |
| People close to me expect me to eat fast food regularly. | 2.86(1.061) |
| **Descriptive *Norms*** | |
| People close to me eat fast food regularly. | 2.84(1.001) |
| The people in my life whose opinions I value eat fast food regularly. | 2.79(1.010) |
| **Perceived Control (α=0.60)** | |
| **Perceived Behavioral Control** | |
| I have the power to control the number of times I will eat fast food during the next month. | 3.73(1.004) |
| The number of times I will eat fast food during the next month is up to my decision at the time. | 3.82(0.849) |
| **Self-efficiency** | |
| It is impossible for me to refrain from eating fast food during the next month. | 3.03(1.200) |
| I can avoid eating fast food regularly during the next month if you want | 3.74(0.936) |
| **Actual behavioral control** | |
| I can't stop myself from eating fast food Because I can't prepare healthy food. | 3.56(1.184) |
| I cannot prevent myself from eating fast food due to being in places that encourage me to buy fast food. | 2.81(1.126) |
| ***Fear of Negative Evaluation* (α=0.86)** | |
| I worry about what others might think of me. | 2.53(1.136) |
| I worry about not being accepted by others. | 2.36(1.173) |
| I worry about what kind of impression I make on others. | 2.82(1.176) |

*(Continued)*

**Table 2.** (Continued)

| Construct items | Mean (SD) |
|---|---|
| I worry that I will say or do the wrong things or that others will notice my shortcomings. | 2.74(1.230) |
| **Self-Identification (α = 0.70)** | |
| I see myself as someone who practices healthy eating habits. | 2.85(0.883) |
| I see myself as someone who is concerned about the health consequences of the foods I eat. | 3.25(0.973) |
| I see myself as someone who is concerned about the environmental consequences of the foods I eat. | 3.07(0.958) |
| I see myself as someone who enjoys eating. | 3.92(0.892) |
| I see myself as Someone interested in eating healthy. | 3.18(1.012) |
| **Intention (α = 0.79)** | |
| Based on my taste preferences and current lifestyle: | |
| I will likely eat fast food regularly over the next month. | 2.72(1.118) |
| I will probably eat fast food regularly over the next month. | 2.75(1.119) |
| My financial ability qualifies me not to abstain from eating fast food. | |

relationship (r = 0.23, P < 0.001) between financial access and frequency of fast-food intake, as well as a direct association (r = 0.24, P < 0.001) between BMI and the amount of fast food consumption.

The Ajzen's TBP model, depicted in Fig 2, failed to effectively explain either the intention to consume or behavior towards fast-food consumption. Despite various modifications, classical theory does not meet all the necessary criteria for a good model fit. Due to its poor fit and space limitations, results were left unreported but revealed inconsistencies between the data collected and the basic TPB theory, which aligns with findings from previous studies, such as [20,61]. Therefore, the extended model includes various factors, such as fear of negative evaluation, self-identification, and interactions between multiple variables, along with the level of income (available income) and BMI. These additional variables were added to the TPB model. However, values with low factor loadings (< 0.5) were removed, along with any nonsignificant predictive variables, to modify the model accordingly for a better fit based on goodness-of-fit measures.

Fig 3 illustrates the extended model, which integrates additional factors identified in the literature. Two external variables, income and BMI, were added as direct predictors of behavior. The model also specifies correlations between affective attitude, cognitive attitude, PBC, ABC, income, and BMI, represented by double-headed arrows. Interestingly, it was found that PBC was not significantly associated with intention but had a significant association with actual behavior. Notably, most PBC questions focused on respondents' perception of their ability to monitor their fast-food consumption frequency rather than their intention to consume fast food. Recent studies have proposed modifications and advancements in the theory, suggesting that PBC plays a crucial role as an intermediary factor between intention and behavior [69].

The study revealed that cognitive attitudes, descriptive norms, and self-identification, in combination, accounted for 7% of the variation in the intention to consume fast food. Affective attitudes were deemed insignificant and did not directly affect intention. Cognitive attitude demonstrated a strong direct association with consumption intention ($\beta = 0.211$, $P < 0.005$), whereas affective attitude had a positive influence on behavior rather than intention ($\beta = 0.130$, $P < 0.045$).

**Table 3. Pearson correlation results.**

|  | Behavior | Intension | Income | BMI | Cognitive Attitude | Affective Attitude |
|---|---|---|---|---|---|---|
| **Behavior** | 1 | .103* | .232** | .241** | .198** | .156** |
| **Intension** | .103* | 1 | 0.038 | 0.017 | .126* | 0.04 |
| **Income** | .232** | 0.038 | 1 | .209** | 0.028 | 0.033 |
| **BMI** | .241** | 0.017 | .209** | 1 | −0.003 | −.115* |
| **Cognitive Attitude** | .198** | .126* | 0.028 | −0.003 | 1 | .502** |
| **Affective Attitude** | .156** | 0.04 | 0.033 | −.115* | .502** | 1 |
| **Inductive Norms** | .134** | .313** | 0.053 | −0.034 | .251** | .263** |
| **Descriptive Norms** | 0.028 | .192** | −0.007 | 0.028 | 0.01 | 0.001 |
| **Subjective Norms** | .111* | .331** | 0.034 | −0.008 | .185** | .188** |
| **Perceived Behavioural Control** | −.307** | 0.039 | −.098* | 0.036 | −.119* | −.179** |
| **Actual control** | −.325** | −.105* | −.165** | −0.012 | −.270** | −.247** |
|  | Inductive Norms | Descriptive Norms | Subjective Norms | Perceived Behavioral Control | Actual control | Self-Identification |
| **Behavior** | .134** | 0.028 | .111* | −.307** | −.325** | −.225** |
| **Intension** | .313** | .192** | .331** | 0.039 | −.105* | .126* |
| **Income** | 0.053 | −0.007 | 0.034 | −.098* | −.165** | 0.004 |
| **BMI** | −0.034 | 0.028 | −0.008 | 0.036 | −0.012 | −0.05 |
| **Cognitive Attitude** | .251** | 0.01 | .185** | −.119* | −.270** | −0.093 |
| **Affective Attitude** | .263** | 0.001 | .188** | −.179** | −.247** | −.116* |
| **Inductive Norms** | 1 | .218** | .836** | −.127* | −.159** | 0.058 |
| **Descriptive Norms** | .218** | 1 | .718** | −0.044 | −0.016 | 0.079 |
| **Subjective Norms** | .836** | .718** | 1 | −.115* | −.122* | 0.086 |
| **Perceived Behavioral Control** | −.127* | −0.044 | −.115* | 1 | .347** | .281** |
| **Actual control** | −.159** | −0.016 | −.122* | .347** | 1 | .106* |

Note: This table presents the Pearson correlation coefficients among key psychological, behavioral, and socioeconomic variables related to fast-food consumption behavior. Values marked with asterisks indicate statistical significance:

*p < 0.10, **p < 0.05, ***p < 0.01.

According to the study's findings, subjective norms were consistently found to have a negative impact on the accuracy of the model and were insignificant in all model adjustments. To improve the predictive ability of the model, descriptive norms were added, resulting in a significant impact of 0.234 on consumption intention (P < 0.000). This suggests that while students may enjoy fast food, their decisions are more influenced by practical considerations rather than emotional ones.

The perceived behavioral control scale demonstrated a direct relationship between behavior and self-control/self-efficacy, which led to lowered intentions towards consumption, with a negative coefficient of −0.37. On the other hand, self-identification had a positive effect on intention (β = 0.204, P < 0.009), while actual control proved to be a significant predictor of behavior towards consumption (β = −0.264, P < 0.000).

The university student behavior model was expanded to include income and BMI as possible influencing factors, and the results indicated a positive impact on the model, with estimates of 0.09 and 0.36 for these variables, respectively. Moreover, students' intentions were positively correlated with their own behavior and accounted for approximately 6% of the variance (P < 0.07).

All covariances were significant at a level less than or equal to 1%. Additionally, it is worth noting that the results revealed a positive association between income levels and BMI in the study sample. The results of the model's fit were deemed satisfactory, and the relative chi-square test yielded significant findings: $x2/df ≈ 3.67$, GFI = .95, AGFI = .91, CFI = .8, RMSEA = .08, PGFI = 0.56, RMR = 0.05.

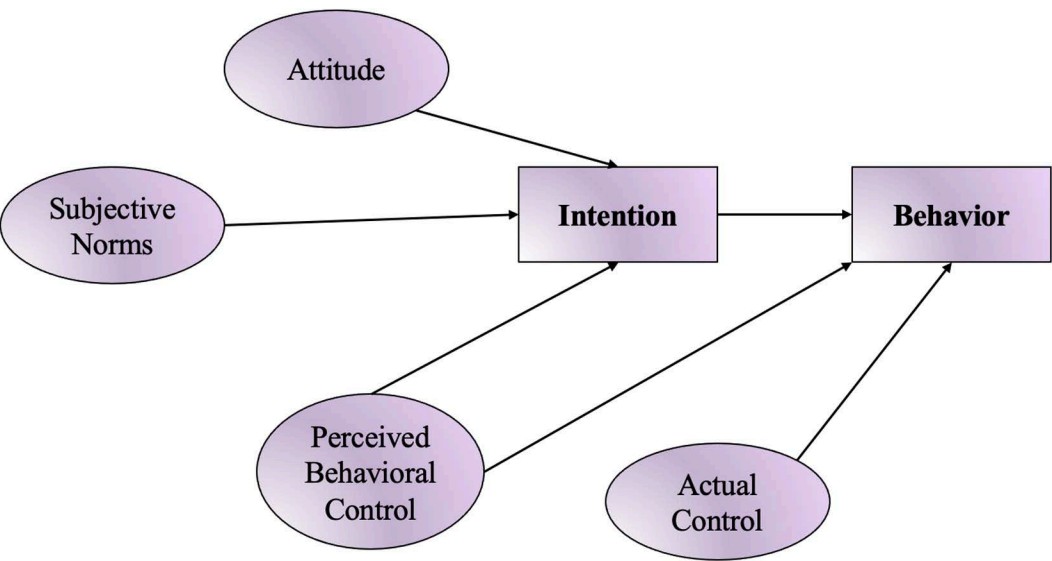

**Fig 2. Path analysis of the classical TPB model, adapted from [ 18].**

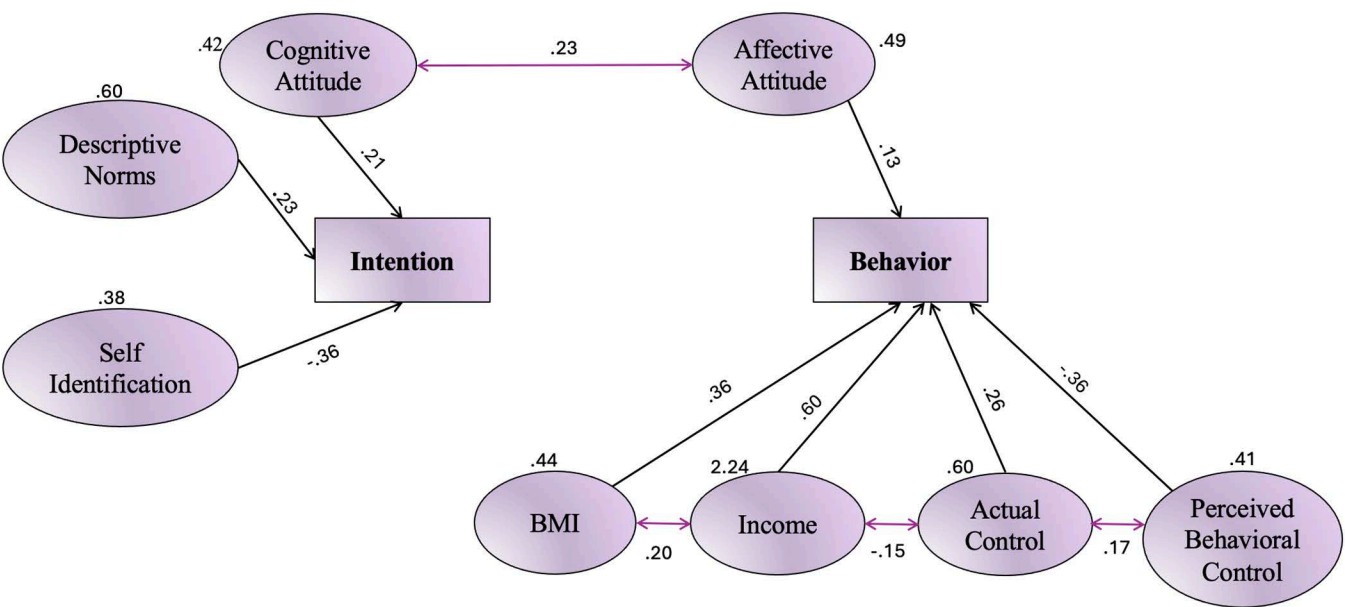

**Fig 3. Path analysis of the extended TPB model predicting fast-food consumption behavior.**

The research revealed that students' intentions served as a moderately significant predictor of their behavior, accounting for 6% of the variance (P < 0.07). Fig 3 shows the extended TPB model, which indicates that both intention and behavior were predicted by the model at 6% and 22%, respectively. However, these percentages are considered low in comparison to other studies that have reported values ranging from 20–50% when it comes to accurately predicting consumption intention. The findings are reported in Table 4.

**Table 4. Summary of Multiple Regression.**

| Construct | Unstandardized Coefficients | | Unstandardized Coefficient | t-value | P-Value | R² |
|---|---|---|---|---|---|---|
| | β | S.E | β | | | |
| **Panel (A) Summary of Multiple Regression: Effect of Intension, Actual control, Affective Attitude, Perceived Behavioral Control, Income, BMI on behavior** | | | | | | |
| *Intension* | 0.078 | 0.045 | 0.077 | 1.75 | 0.08 | 0.22 |
| *Actual control* | −0.264 | 0.063 | −0.196 | −4.188 | *** | |
| *Income* | 0.086 | 0.031 | 0.125 | 2.761 | *** | |
| *BMI* | 0.366 | 0.07 | 0.233 | 5.206 | *** | |
| *Affective Attitude* | 0.13 | 0.065 | 0.088 | 2.004 | 0.045 | |
| *Perceived Behavioral Control* | −0.365 | 0.075 | −0.225 | −4.842 | *** | |
| **Panel (b) Summary of Multiple Regression: Effect of Descriptive Norms, Cognitive Attitude, Self-Identification on intension** | | | | | | |
| *Descriptive Norms* | 0.227 | 0.063 | 0.174 | 3.626 | *** | 0.07 |
| *Cognitive Attitude* | 0.206 | 0.076 | 0.131 | 2.722 | *** | |
| *Self-Identification* | −0.21 | 0.079 | 0.127 | 2.65 | *** | |
| **Panel (c) Summary of covariances: intension** | | | | | | |
| *Affective <-->Cognitive* | 0.228 | 0.025 | | | *** | |
| *PBC <-->ABC* | 0.165 | 0.026 | | | *** | |
| *Income <-->ABC* | −0.148 | 0.053 | | | *** | |
| *Income <-->BMI* | 0.204 | 0.05 | | | *** | |

It appears that the self-reported data used to evaluate intention-to-behavior were underestimated. As a result, an indirect assessment of intention measurement was constructed to accurately determine actual consumption intentions while mitigating any potential social desirability bias (see, for example, [70,71]). "In accordance with [20], participants systematically underestimated their consumption of fast food. Moreover, their responses regarding the frequency of consumption were significantly lower than their actual consumption. Two methods were employed to assess their behavior: retrospective and prospective. The results indicated that the prospective measure, which entailed recording the actual consumption in a daily diary, yielded more accurate outcomes. This finding implies that individuals may tend to underestimate dietary risks while overestimating their intake of healthy foods [72,73]." Thus, the revised projective measure of intention comprises six factors that enable us to indirectly gauge individuals' propensity to consume fast food. The modified construct assesses students' perception of their emotional and cognitive ability to eat fast food, their ability to pay for fast food meals, recognition of the risk associated with consumption, lack of guilt and worry after consumption, and concern for health and environmental repercussions stemming from the behavior.

The self-reported and structured projective measures were compared using paired-sample t-tests. The results demonstrated that participants underestimated their fast-food consumption intention, as their self-reported intention consumption (M = 2.70, SD = 1.00) was significantly lower than that of the projective measure (M = 3.20, SD = 0.565; t = 7.52, p < .001).

The extended TPB model, which includes the new construct of intention (projective) illustrated in Fig 4, was more effective in explaining behavioral variations than the previous model. The newly used construct revealed that intention had a positive and statistically significant influence (0.21; p < 0.001) on behavior and explained 22% of its variation. Notably, attitude, descriptive norms, and self-identification accounted for 54% of the variation in the intention to consume fast food. "FNE did not contribute substantially to the variation in either intention or behavior, or its interactive effect with Subjective Norms. Consequently, it was omitted from the final model. This conclusion aligns with the results reported by [20]".

This study found that the intention to eat fast food was determined by attitude (with a direct effect), subjective norms (with a direct effect), and self-identification (indirect effect). The results showed that attitude was the primary predictor of fast-food consumption intentions, followed by self-identification and descriptive norms.

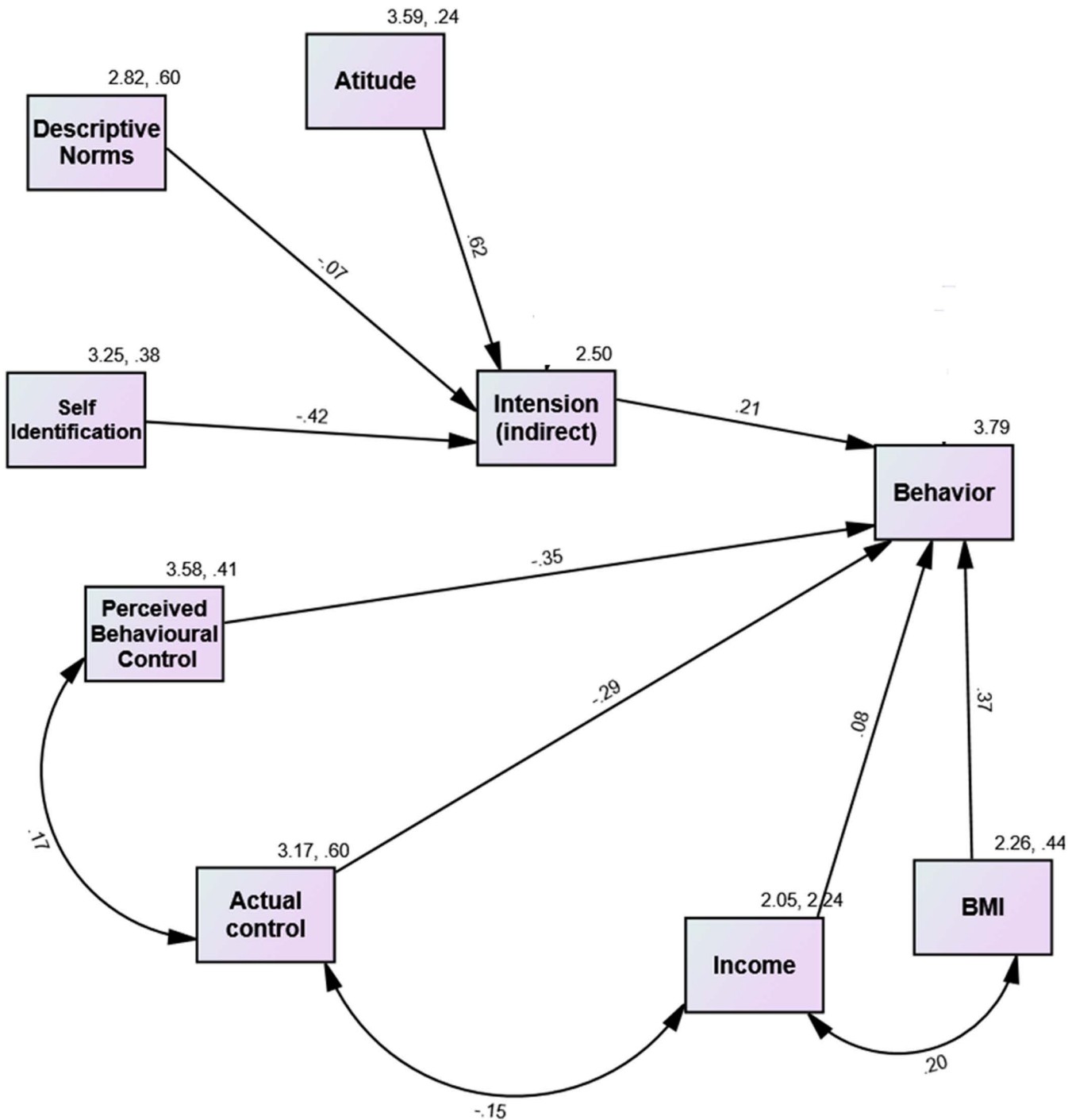

**Fig 4. Path analysis of the extended TPB model using projective (indirect) intention to predict fast-food consumption behavior.** The diagram presents standardized beta coefficients for direct paths. Double-headed arrows indicate correlations among PBC, ABC, income, and BMI. All relationships depicted are statistically significant.

As expected, identifying a healthy eater was associated with a decrease in the intention to consume fast food (β = −0.42, P < 0.000). Both cognitive and affective attitudes were found to have the greatest impact on explaining intention (β = 0.623, P < 0.000), while descriptive norms had a negative effect. The study further revealed that social influence plays an important role in fast-food consumption; strong subjective norms lead to a decreased perception of control overeating fast-food.

Similar to previous models, PBC and ABC had a significant negative impact on behavior. The study found that actual control was strongly linked to behavior (with a β coefficient of −0.264 and a P value less than zero). The findings suggest strong correlations between IBM-income, PBC-actual control, and income-actual control. Specifically, a positive correlation exists between IBM-income, and a negative correlation between income and actual control. These associations are highly significant. The results are summarized in Table 5. This model provided a good fit with a significant relative chi-square test (x2/df = 3.67, GFI = .96, CFI = .87, RMSEA = .08).

## 5. Discussion

This study aimed to explore the determinants of fast-food consumption among university students in Jordan using an extended version of the Theory of Planned Behavior. The findings provide valuable insights into the factors influencing fast-food consumption behaviors and offer significant implications for future research and practical applications.

The results highlight the critical role of cognitive attitudes in predicting the intention to consume fast food. Students' beliefs about the convenience, taste, and enjoyment of fast food were found to significantly shape their consumption behaviors. This underscores the need to target these beliefs in interventions aimed at promoting healthier eating habits. Additionally, the influence of subjective norms and self-identification on fast-food consumption intentions suggests that social-based interventions could be effective. Interestingly, this study found that descriptive norms were more significant predictors of fast-food consumption intention than injunctive norms. This suggests that students are more influenced by what they observe their peers doing (e.g., eating fast food frequently) than by whether those behaviors are socially approved or disapproved. This finding aligns with recent extensions of the TPB, which emphasize that descriptive norms can be powerful [74] in environments where behaviors are highly visible and socially embedded, such as university campuses [75]. The distinction is important, as it suggests that public health interventions may be more effective when

**Table 5. Summary of Multiple Regression.**

| Construct | Unstandardized Coefficients | | Unstandardized Coefficient | t-value | P-Value | R² |
|---|---|---|---|---|---|---|
| | β | S.E | β | | | |
| **Panel (A) Summary of Multiple Regression: Effect of Intension, Actual control, Perceived Behavioral Control, Income, BMI on behavior** | | | | | | |
| *Intension2(indirect)* | *0.211* | *0.083* | *0.111* | *2.547* | *0.01* | *0.22* |
| *Actual control* | *−0.290* | *0.063* | *−0.215* | *−4.591* | *\*\*\** | |
| *Income* | *0.084* | *0.031* | *0.121* | *2.694* | *\*\*\** | |
| *BMI* | *0.369* | *0.07* | *0.235* | *5.257* | *\*\*\** | |
| *Perceived Behavioral Control* | *−0.349* | *0.075* | *−0.215* | *−4.591* | *\*\*\** | |
| **Panel (b) Summary of Multiple Regression: Effect of Attitude, Descriptive Norms, Self-Identification on the Intension (Indirect).** | | | | | | |
| *Attitude* | *0.623* | *0.038* | *0.554* | *16.451* | *\*\*\** | *0.54* |
| *Descriptive Norms* | *−0.068* | *0.024* | *−0.097* | *−2.88* | *\*\*\** | |
| *Self-Identification* | *−0.419* | *0.03* | *−0.471* | *−13.973* | *\*\*\** | |
| **Panel (c) Summary of covariances: intension (indirect)** | | | | | | |
| *Income <--> BMI* | *0.204* | *0.05* | | | *\*\*\** | |
| *PBC <--> ABC* | *0.165* | *0.026* | | | *\*\*\** | |
| *Income <--> ABC* | *−0.148* | *0.053* | | | *\*\*\** | |

they target perceived norms of behavior, rather than solely focusing on approval or moral messaging. Thus, encouraging healthy eating behaviors within peer groups and strengthening students' health-conscious identities could help reduce unhealthy eating habits.

Building on these psychosocial determinants, demographic and physiological factors also emerged as significant predictors. The study revealed a positive correlation between higher income (personal funds), BMI, and frequent fast-food consumption. This indicates that financial flexibility and weight-related factors significantly influence dietary choices. Addressing the role of financial flexibility and providing affordable healthy food options could mitigate the impact of fast-food consumption, particularly among higher-income students. While this finding appears to contradict findings from high-income countries, where lower income is generally associated with higher obesity due to limited access to nutritious food [76], this relationship may not hold in the Jordanian context. In fact, local evidence suggests the opposite. A study by [77] found that Jordanian school children from higher-income families were more likely to be obese, and [78] showed that university students with higher daily pocket money had significantly greater odds of frequent fast-food consumption and elevated BMI. These findings suggest that in Jordan, greater financial flexibility may facilitate access to calorie-dense convenience foods, particularly among youth, reinforcing a positive rather than negative relationship between income and obesity.

While income may shape access to fast food, individual-level determinants remain crucial. In this regard, perceived and actual behavioral control emerged as significant predictors of consumption behavior. Students with higher perceived control over their eating habits and actual control over their food environment were less likely to consume fast food. Enhancing students' self-efficacy and providing supportive environments can effectively reduce fast-food intake. These findings suggest that dietary choices are shaped by a complex interplay of factors, including economic access, BMI, social identity, and environmental cues. While higher available income and BMI were associated with more frequent fast-food consumption in this sample, these variables do not operate in isolation. Psychological drivers such as stress and convenience, along with social influences like peer eating habits and food marketing, also contribute meaningfully to dietary behavior. Accordingly, addressing fast-food consumption patterns among university students may require more than just improving access to affordable healthy foods; it also demands strategies that consider social dynamics, perceived norms, and the broader food environment that students navigate daily.

Although this study did not examine directly whether students consume fast food alone or with others, it is likely that such behavior often serves a social function. Fast-food outlets may act as informal gathering spaces for students to socialize, relax, or connect with peers. In such cases, food choices may be shaped less by individual preference and more by group dynamics and perceived social norms. Merely emphasizing the health risks of fast food may therefore be insufficient to change behavior. Effective interventions should also address the social motivations behind fast-food consumption, for example, by promoting healthy options in communal settings, reframing peer norms, and embedding behavioral cues in group dining environments. Future research should investigate the social contexts of fast-food consumption in more depth, such as whether students eat alone or with peers, and how social norms and group dynamics influence their dietary decisions.

Beyond these social influences, the findings also revealed a psychological discrepancy between students' reported intentions and their underlying behavioral tendencies. Specifically, students tended to underestimate their intention to consume fast food, suggesting a potential gap between what they believe or wish to report and their actual inclinations. This discrepancy underscores the need for more accurate and less biased measurement tools, such as projection-based assessments, to better capture true behavioral intent. Relying solely on self-reported data for attitudes, norms, and consumption behavior may introduce social desirability bias and misreporting. Future studies should consider incorporating objective measures and observational data to validate and triangulate self-reported findings.

However, it is essential to acknowledge the study's limitations. The cross-sectional nature of the study limits the ability to establish causal relationships between the variables. Longitudinal studies are needed to better understand the temporal

dynamics of fast-food consumption behaviors. The study sample was limited to university students in Jordan, which may not be representative of the broader population. Replicating the study in different settings and among diverse demographic groups would enhance the generalizability of the findings. Furthermore, the study focused on specific predictors within the TPB framework and excluded other potential factors such as emotional eating, stress levels, and cultural influences. Future research is needed to consider a more comprehensive approach to capture the multifaceted nature of fast-food consumption behaviors.

In conclusion, the findings from this study contribute to the understanding of fast-food consumption behaviors among young adults in Jordan. By identifying key psychosocial and economic determinants, the study offers valuable insights for designing effective interventions to promote healthier eating habits. Additionally, by revealing why fast-food consumption is rising among students, the findings lay the groundwork for developing targeted strategies that address both psychosocial and structural influences. Future research should build on these findings, addressing the limitations and exploring additional factors to develop a more holistic understanding of fast-food consumption behaviors.

## 6. Conclusions

This study set out to explore the key behavioral and contextual factors that shape fast-food consumption among university students in Jordan. To better understand what drives these choices, we applied an extended version of the Theory of Planned Behavior, incorporating additional variables such BMI and income (personal funds). Through two separate factor analyses, we examined how elements like attitudes, social norms, perceived and actual control, and self-identification relate to students' intentions to consume fast food, and how those intentions translate into actual eating behavior.

The study found that students often underestimated their intention to consume fast food. When a structured projective measurement was applied, a much stronger and statistically significant link emerged between intention and actual behavior. One of the key findings was that intention plays an important role in shaping consumption, explaining 22% of the variation in fast-food intake. The results also reinforced the relevance of belief-based factors within the Theory of Planned Behavior: attitudes, subjective norms, and self-identification together accounted for 53% of the variation in intention. Among these, attitudes, particularly emotional responses, had the strongest influence on students' intentions, highlighting the central role of affect in motivating fast-food consumption. The impact of marketing strategies and visible peer behaviors likely contributed to these affective and normative influences, making fast food appear more socially acceptable, convenient, and rewarding. These insights suggest that promoting healthier food options on campus, while also addressing the social and marketing cues that shape eating norms, could help reduce students' reliance on fast food. Strengthening a healthy-eater identity may further shift attitudes and reduce intention to consume unhealthy food.

The study also found that both perceived and actual behavioral control were significantly linked to lower fast-food consumption, suggesting that students who feel more in control of their eating habits are less likely to rely on fast food. In addition, a notable association emerged between higher income, higher BMI, and more frequent fast-food intake. This indicates that fast-food consumption is not limited to lower-income groups, but is also common among students with greater financial means. These findings underscore the need to explore how both financial limitations and increased affordability can shape unhealthy eating patterns. Designing targeted interventions that consider both psychological drivers and financial flexibility could help reduce fast-food consumption among students. Future research should explore strategies that shift students' motivation and perception, especially in cultural contexts like Jordan, where fast food is becoming increasingly normalized.

## Supporting information

**S1 Dataset.  Minimal data set containing coded variables and values used in all statistical analyses, including those used to generate figures and regression models.**
(XLSX)

**S1 Data. De-identified Arabic-language questionnaire responses collected during the study.** Provided for reference and contextual transparency.

(XLSX)

## Author contributions

**Conceptualization:** Noura Abu Asab, Randah Barakat.

**Data curation:** Noura Abu Asab.

**Formal analysis:** Noura Abu Asab.

**Investigation:** Noura Abu Asab.

**Methodology:** Noura Abu Asab, Randah Barakat.

**Project administration:** Noura Abu Asab.

**Software:** Randah Barakat.

**Supervision:** Noura Abu Asab.

**Validation:** Randah Barakat.

**Writing – original draft:** Noura Abu Asab.

**Writing – review & editing:** Noura Abu Asab, Randah Barakat.

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
