## [Decision Letter · Decision Letter 0]

Dear Dr. Abu Asab,

Thank you for submitting your manuscript to PLOS ONE. After careful consideration, we feel that it has merit but does not fully meet PLOS ONE’s publication criteria as it currently stands. Therefore, we invite you to submit a revised version of the manuscript that addresses the points raised during the review process.

We look forward to receiving your revised manuscript.

Kind regards,

António Raposo

Academic Editor

PLOS ONE

Journal Requirements:

“The authors have declared that no competing interests exist.The authors certify that they have NO affiliations with or involvement in any organization or entity with any financial interest (such as honoraria; educational grants; participation in speakers’ bureaus; membership, employment, consultancies, stock ownership, or other equity interest; and expert testimony or patent-licensing arrangements), or non-financial interest (such as personal or professional relationships, affiliations, knowledge or beliefs) in the subject matter or materials discussed in this manuscript.”

3. In the online submission form, you indicated that [The data underlying the results presented in the study are available from the authors upon request. Due to ethical considerations, the data cannot be shared publicly to protect the confidentiality and privacy of the participants involved. Interested researchers can contact the corresponding author to obtain access to the data, provided they agree to comply with the ethical guidelines and data protection protocols established by the study's review board.].

Reviewers' comments:

Reviewer's Responses to Questions

**Comments to the Author**

1. Is the manuscript technically sound, and do the data support the conclusions?

Reviewer #1: Yes

Reviewer #2: Partly

Reviewer #3: No

2. Has the statistical analysis been performed appropriately and rigorously?

Reviewer #1: Yes

Reviewer #2: Yes

Reviewer #3: Yes

3. Have the authors made all data underlying the findings in their manuscript fully available?

Reviewer #1: Yes

Reviewer #2: Yes

Reviewer #3: Yes

4. Is the manuscript presented in an intelligible fashion and written in standard English?

Reviewer #1: Yes

Reviewer #2: No

Reviewer #3: Yes

Reviewer #1: Dear Author,

Thank you for your valuable work. Below, I have provided some insights to help improve the

paper:

1.

Methods Section:

I recommend including a graph that illustrates the key variable of interest, based on the

theoretical framework, in the Methods section. This would enhance the readers'

understanding of your approach and improve the overall readability of the paper.

I noticed that you defined the variable using the question format described in the paper cited

at reference 20. I kindly invite you to elaborate on why a validated questionnaire was not

utilized to investigate this area. Providing such insights would strengthen the clarity and

robustness of your methodological approach.

2.

Figures 1 and 2: Consider redesigning Figures 1 and 2 to make them more visually appealing

and clearer. A more polished design could increase their impact and effectiveness in

conveying the intended message.

3.

Discussion Section: You stated, "Moreover, the study revealed a positive correlation between

higher income, BMI, and frequent fast-food consumption." However, this finding contrasts

with the existing literature, which generally suggests that higher BMI is associated with lower

income. I suggest including references that establish this general trend, as well as studies

specific to Jordan, to explore whether your findings reflect regional nuances.

4.

Income Variable: Since your study focuses on the student population, the use of "income" as

a variable requires clarification. In this context, "income" likely refers to family income, which

may not accurately represent students’ financial independence or spending habits.

Furthermore, students often manage money that they do not directly earn, which limits the

variable's relevance. I suggest reconsidering the role of income in your analysis or reframing

its significance in your discussion.

5.

Economic Status and BMI: The expression, "This indicates that economic status and weight

related factors significantly influence dietary choices," could be improved to reflect a more

nuanced interpretation. Additionally, the sentence, "Addressing these economic barriers and

providing affordable healthy food options could mitigate the impact of fast-food

consumption, particularly among higher-income students," should consider broader social

factors.

6.

Social Context of Fast-Food Consumption: Investigate whether fast-food consumption

among students serves as a form of socialization or if students visit fast-food outlets alone.

Understanding these social dynamics is crucial for devising strategies to reduce fast-food

consumption. Merely emphasizing the health risks of fast food may not suffice to change

behavior. Instead, addressing the social motivations behind fast-food visits could provide

more effective interventions.

I hope these suggestions help improve your manuscript.

Reviewer #2: Dear Editor in Chief

Plos ONE

The paper: Analysing Fast Food Consumption Behaviour Patterns: The Case of Jordan is a good study and a worthy research topic. It is quiet relevant to Plos One topics. However, the article cannot be considered for publication in Plos One as it is. It needs a strong revision, as there are serious readability problems and methodological issues that the author(s) must manage effectively. I offer some questions and comments with the hope of helping s/he/them improve their work. I wish the research team the very best of luck as they continue work in this domain.

Overall evaluation.

It is a worthy research topic and a interesting study but the paper needs a strong revision.

I hope that the comments are fair and constructive. All Best!

Comment 1) Introduction” section needs revision. The author(s) must underline the initial assumptions of the paper and the originality of the paper – contribution to knowledge. I would suggest considering the following five (5) basic elements in the section of introduction as subheadings of Introduction:

a) Research aim:

b) Initial assumptions of the paper

c) Reasoning for the focus of the paper,

d) Research objectives,

e) Originality of the paper and contribution to knowledge.

The author (s) tried to write based on above, but unfortunately is so poor.

In the last paragraph of the Introduction, the authors should clearly mention the weaker point of former works (identification of the gaps) and describe the novelties of the current investigation to justify the paper deserves to be published in food marketing journals. For example, it is not explained about stage one at section aim in the last paragraph of Introduction. Section one is missing.

What is TPB in section Introduction. Mention the complete form of abbreviation when it is included at the first time.

Comment 2) you have to add Background to your paper.

Comment 3) Method and results Section needs revision. The description of this is tedious and it’s so hard for reader to understand. It is mentioned 408 participants in section abstract. However it is written 400 in this section. There are discrepancies in these parts. Correct it.

Comment 4) Research results: It is so long and it is needed to shorten. It is like a report. There are lots of hypothesis that is not required at this part. In section “Theorical…”, it is mention some number as reference instead of the name of authors like [17], [18], [32], [36,37,38,39..].

Comment 5) Discussion section. Author(s) must explain what is the different of his study with others and if they are same, why they are similar? Comparison must be with papers after 2013. It needs to expand the importance of findings. No study is included at this part. It is like a report.

Comment 6) Conclusion is so poor and so long. Shorten and present the main finding of study.

Comment 7) it is important to review the article’ writing. There are some typos throughout the text.

Reviewer #3: This article aims to investigate the factors influencing fast food consumption among university students in Jordan. However, it does not clearly explain why this particular group needs to be focused on or the reasons behind this choice. Additionally, the title and abstract should be revised to reflect that this study is on university students. the literature emphasizes the Theory of Planned Behavior (TPB), yet there is a lack of literature review relating specifically to the content of this study.

Moreover, the sample is entirely drawn from the University of Jordan. The rationale for selecting this institution as the sole source of participants is not provided. Can this university truly represent the entire university student population of Jordan? Furthermore, 75% of the sample are female students. Does this proportion align with the actual gender distribution among university students in Jordan? How can the authors demonstrate that this constitutes a valid and representative sample?

**Do you want your identity to be public for this peer review?** For information about this choice, including consent withdrawal, please see our Privacy Policy

Reviewer #1: No

Reviewer #2: No

Reviewer #3: No

---

## [Author Response · Author response to Decision Letter 1]

12 May 2025

Response to Reviewer Comments

Manuscript: Analysing Fast Food Consumption Behaviour Patterns: The Case of Jordan

• Comment 1: Use of Non-Validated Questionnaire

We thank the reviewer for this important observation. While validated instruments offer consistency, we deliberately tailored our questionnaire to align with the behavioral, linguistic, and cultural context of Jordanian university students. The instrument retained the core structure of the Theory of Planned Behavior, adapting the extended theory constructs used by Dunn et al. (2011) to enhance relevance and clarity for the target population.

The adapted questionnaire was pilot-tested, revised based on feedback, and assessed for internal consistency (Cronbach’s alpha: 0.60–0.86). This approach is consistent with prior Theory of Planned Behavior-based research in non-Western settings.

To support this methodological framework, we have also added a graph in the Methods section (now Figure 1) that visually illustrates the study’s conceptual framework, based on the extended Theory of Planned Behavior, and key variables of interest. This addition improves reader comprehension of the theoretical structure guiding our analysis.

• Comment 2: Figures 1 and 2 - Clarity and Design

We appreciate the suggestion. Figures 1 and 2 have been redesigned and renamed as Figures 2 and 3, while a new Figure 1 was added in the Methods section to illustrate the full conceptual framework. The revised figures use clearer layoutsand improved labeling to enhance readability and impact.

• Comment 3: BMI- Income

Thank you for this insightful comment. We now reference Burgoine et al. (2018) to reflect global trends showing a negative relationship between income and BMI in high-income settings. To contextualize our contrasting result, we included Jordan-specific studies (Khader et al., 2009; Khatatbeh et al., 2022) showing that students with higher income tend to consume more fast food and exhibit higher BMI levels, likely due to greater access to calorie-dense options.

• Comment 4: Clarification of 'Income' Variable

We agree with the reviewer. The term “income” was clarified in the Methods section as referring to students’ available personal funds, including family support, stipends, part-time jobs, and scholarships. It serves as a proxy for financial flexibility rather than independence.

• Comment 5: Economic Status and BMI Interpretation

Thank you for this thoughtful comment. We agree that the original phrasing could be refined to better reflect the complexity of factors influencing dietary choices. In response, we have revised this section of the discussion to emphasize that dietary behavior is shaped by a multidimensional interplay of economic status, physiological characteristics (e.g., BMI), psychological dispositions, and social-environmental influences such as food marketing, peer behavior, and cultural norms. We also revised the second sentence to reflect that effective interventions should not only address affordability and access but also tackle social motivations and environmental cues that shape food choices among students. These revisions now provide a more comprehensive and context-sensitive interpretation of the findings.

• Comment 6: Social Context of Fast-Food Consumption

We fully agree. While our study did not explicitly assess whether students consumed fast food alone or in social groups, we agree that the social context of consumption is a critical factor in understanding food choices among young adults. In response, we have revised the Discussion section to acknowledge the likely social function of fast-food consumption, such as peer bonding, group outings, or convenience in communal settings. We also note that social motivations and norms may significantly influence dietary behavior, potentially limiting the effectiveness of health messaging alone. The revised text emphasizes the need for interventions that consider social dynamics, and we recommend that future research explore whether students eat alone or with peers, and how group behaviors shape consumption decisions.

Additional comments:

• Comment: Methods and Results - Clarity and Consistency

Thank you for the feedback. The Methods and Results sections have been revised for clarity and conciseness. To address the discrepancy, we clarified that 408 students participated and ensured consistency across the abstract and body of the paper. The phrasing "around 400" in the methodology reflects the initial planning stage; the final figure of 408 is used consistently in the Results and Abstract.

• Comment: Language and Typographical Errors

We appreciate this note. The entire manuscript has undergone a thorough language and grammar review. Typos and stylistic inconsistencies have been corrected to improve clarity and ensure professional quality.

• Comment: Sampling and Representativeness

We acknowledge this important point. In response, we have clarified in the Methods section that the University of Jordan was selected due to its size, diversity of academic programs, and geographic centrality. These features make it a suitable and meaningful cross-section for studying university students in Jordan. While the sample may not fully represent all students nationwide, its demographic diversity provides strong grounds for initial exploration. We also addressed the gender imbalance in the sample, noting that the high proportion of female participants (approximately 75%) aligns with national enrollment patterns in Jordanian universities, particularly within the humanities and social sciences. This clarification supports the interpretability and relevance of the findings within the context of the local higher education landscape.

• Comment: The Introduction section needs revision.

The Introduction has been thoroughly revised to include a clear research aim, theoretical assumptions, justification for the population focus, explicit objectives, and a statement of the study’s originality and contribution to the field.

We thank the editor and reviewers for their valuable comments, which significantly improved the clarity and rigor of this paper.

---

## [Decision Letter · Decision Letter 1]

Analysing Fast Food Consumption Behaviour Patterns: The Case of Jordan

PONE-D-24-31310R1

Dear Dr. Abu Asab,

We’re pleased to inform you that your manuscript has been judged scientifically suitable for publication and will be formally accepted for publication once it meets all outstanding technical requirements.

Kind regards,

António Raposo

Academic Editor

PLOS ONE

Additional Editor Comments (optional):

Reviewers' comments:

Reviewer's Responses to Questions

**Comments to the Author**

Reviewer #1: All comments have been addressed

2. Is the manuscript technically sound, and do the data support the conclusions?

Reviewer #1: Yes

3. Has the statistical analysis been performed appropriately and rigorously?

Reviewer #1: Yes

4. Have the authors made all data underlying the findings in their manuscript fully available?

Reviewer #1: Yes

5. Is the manuscript presented in an intelligible fashion and written in standard English?

Reviewer #1: Yes

Reviewer #1: After the revision, the article appears clearer and more understandable. It is stronger and more consistent overall. All sections are well explained, and it offers new insights into students' food habits, providing a focused perspective through the case of Jourdan.

**Do you want your identity to be public for this peer review?** For information about this choice, including consent withdrawal, please see our Privacy Policy

Reviewer #1: No

---

## [Editor Report · Acceptance letter]

PONE-D-24-31310R1

PLOS ONE

Dear Dr. Abu Asab,

I'm pleased to inform you that your manuscript has been deemed suitable for publication in PLOS ONE. Congratulations! Your manuscript is now being handed over to our production team.

Kind regards,

on behalf of

Dr. António Raposo

Academic Editor

PLOS ONE